# Injection of Raffinose Family Oligosaccharides at 12 Days of Egg Incubation Modulates the Gut Development and Resistance to Opportunistic Pathogens in Broiler Chickens

**DOI:** 10.3390/ani10040592

**Published:** 2020-03-31

**Authors:** Katarzyna Stadnicka, Joanna Bogucka, Magdalena Stanek, Radomir Graczyk, Krzysztof Krajewski, Giuseppe Maiorano, Marek Bednarczyk

**Affiliations:** 1Department of Animal Biotechnology and Genetics, Faculty of Animal Breeding and Biology, UTP University of Science and Technology, 28 Mazowiecka, 85-022 Bydgoszcz, Poland; marbed13@op.pl; 2Department of Animal Physiology, Physiotherapy and Nutrition, Faculty of Animal Breeding and Biology, UTP University of Science and Technology, 28 Mazowiecka, 85-022 Bydgoszcz, Poland; bogucka@utp.edu.pl (J.B.); Magdalena.Stanek@utp.edu.pl (M.S.); 3Department of Biology and Animal Environment, Faculty of Animal Breeding and Biology, UTP University of Science and Technology, 28 Mazowiecka, 85-022 Bydgoszcz, Poland; Radomir.Graczyk@utp.edu.pl; 4Vetdiagnostica, Accredited Veterinary Diagnostic Laboratory Unit, Otorowo 30, 86-050 Solec Kujawski, Poland; krzysztof@vetdiagnostica.pl; 5Department of Agricultural, Environmental and Food Sciences, University of Molise, 86100 Campobasso, Italy; maior@unimol.it

**Keywords:** in ovo, prebiotic, gut health, raffinose family oligosaccharides, broiler, opportunistic pathogens

## Abstract

**Simple Summary:**

In the face of a changing climate, antibiotic resistance and uprising outbreaks of ‘forgotten’ diseases, there is an urgent need for new, safe strategies and natural immunomodulatory products in intensive broiler production. So far, many prebiotic and synbiotic preparations have been explored to influence the gut microbiota composition and the host immune system. However, the effects of bioactive compounds are not always found to be consistent. Global analysis allows us to define at least several reasons for those discrepancies: different chemical composition and origins of the oligosaccharides, interaction with other feed ingredients, and unfavorable environmental impact, where the two latter seem to be most important. The in ovo strategy to automatically inject prebiotics at day 12 of egg incubation has been elaborated to eliminate some of those factors and provide a fully controlled modulation of the host biology. Here, the prebiotic modulatory effects were reflected in the improved gut structure and resistance to opportunistic pathogens in the final weeks of broiler rearing, showing lifelong effects. The in ovo strategy allows for the earliest possible immunomodulatory treatments with the use of naturally sourced bioactive compounds, one of them being legume raffinose oligosaccharides.

**Abstract:**

The aim was to investigate the impact of an automatic in ovo injection of the raffinose family oligosaccharides (RFO) extracted from the seeds of *Lupinus luteus* L, on the chicken performance and resistance in a production environment. At day 12 of incubation, a total of 57,900 eggs (Ross 308) were divided into two groups: 1/ Control, injected with 0.9% NaCl and 2/ RFO group, injected with 1.9 mg/egg of the lupin seed extract, dissolved in 0.2 mL NaCl. The performance parameters, biochemical indices (lipid profile, hepatic parameters), gut histomorphology and duodenum structure, oxidative stability of the meat and microbiological counts of the major commensal microbiota species were analyzed. Mortality, body weight, and feed conversion ratio (*FCR*) were not affected. By day 42, several health indices were improved with RFO and were reflected in a beneficial lipid blood profile, increased villi surface and better combating opportunistic pathogens through reduction of *Clostridia* and decreased coccidia counts. The RFO increased meat oxidation, but only at the beginning of the storage. The RFO sourced from local legumes can be considered a promising prebiotic for broiler chickens. In ovo delivery of prebiotics and/or synbiotics should be further optimized as an important strategy for the earliest possible modulation of chicken resistance.

## 1. Introduction

In the era of antibiotic resistance, a need for efficient immunomodulators has urgently increased. From the scientific point of view, the lifelong immunomodulatory effect of a single dose of a prebiotic or a synbiotic compound delivered on day 12 of egg incubation cannot be neglected and this study model can be potentially expanded to an unlimited number of emerging bioactive substances. From our previous studies, we have learned that in order to draw and understand the broader image of the effects of a bioactive compound on the host, a multi-direction biological analyses must be provided [1,2,3,4,5]. To date, only few in ovo studies reported the application of prebiotics in ovo [6,7]. Regarding the performance objectives (body weight, feed intake and feed conversion ratio, *FCR*), the results for different prebiotics or synbiotics are found to be inconsistent, but consequently supporting the stimulating effect on gut development [8], immune gene expression [9], and improved resistance [10]. 

Raffinose is considered an emerging prebiotic [11] and the raffinose family oligosaccharides (RFO) are low molecular weight plant-storage carbohydrates, naturally occurring in legumes [12]. The RFO prebiotic can be extracted from the seeds of a lupin, an undervalued legume which has been recently considered an alternative protein source for feeding monogastric species. 

Previously, we tested RFO injected in ovo at day 12 of egg incubation in a field trial, in several broiler breeds—Cobb, Ross and Hubbard. The increased body weight, but also the increased *FCR* were observed and the bacteria counts showed elevated levels for *Bifidobacteria* species after hatching [13]. The in ovo delivery of other natural-derived prebiotics: galactooligosaccharides synthesized from milk lactose, or algae laminarin and fucoidans, was associated with improvements in a number of parameters of relevance to commercial poultry production [14]. 

In this study, we provided new evidence from the gut histomorphometry and serum indices and we took into account the opportunistic pathogenic stressors, to show the host systems response to the prebiotic stimulus in ovo. We counted the levels of major commensal bacteria groups (*Enterobacteriaceae, Lactobacillaceae* and *Bifidobacteriaceae*) and parasitic oocysts in feces to reflect the physiological and health status of the observed productive performance. The experiments were designed on a large cohort of broiler chickens in a field validation study. The active prebiotic component was obtained from the local lupin and delivered in ovo using the automatic egg injection machine at the earliest possible moment pre-hatch, which is at day 12 of egg incubation. 

The observed lifelong phenotypes lead us to further hypotheses about the epigenetic changes that may initiate the post-generation effects through nutrition and through in ovo microbiome programming. The microbiome constitutes an important part of a host genome. Therefore the in ovo treatments that directly influence the microbial communities can be considered ‘manipulations’ on genes inside the host. In a discussion, we provide the rationales for implementation of the possibly earliest, pre-hatch strategies, to modulate resistance through the host microbiome and gut health.

## 2. Materials and Methods

### 2.1. Egg Incubation and In Ovo Injections

The study was conducted using Ross 308 breeders eggs, incubated in a hatchery (Drobex-Agro Sp. z o.o., Solec Kujawski, Poland). A total of 57,900 eggs was randomly distributed to Petersime incubators (Petersime NV, Zulte, Belgium). The experiments were performed in three runs. At each experimental run, the eggs were randomly allotted into two experimental groups: RFO and Control. At day 12 of egg incubation, the eggs containing viable embryos were automatically injected into the egg air chamber using the improved prototype of the automatic injecting device [13]. The injections were performed with efficacy of up to 30,000 injections/ per hour, including the operation of a post-injection hole sealing system (Appendix A). In the RFO group, each egg was injected with the prebiotic at a dose of 1.9 mg/egg dissolved in 0.2 mL of 0.9% physiological saline optimized earlier by Bednarczyk et al. [13] and the Control group was injected with 0.2 mL of 0.9% physiological saline only. The rationale to deliver prebiotic in ovo precisely on day 12 of egg incubation was based on a series of experiments reported elsewhere [15]. The RFO prebiotic was extracted from the *Lupinus luteus* L. cv. Lord seeds [16]. 

### 2.2. Rearing Management and Performance Data

In total, 49,000 broiler chickens were subject to the rearing after injections in ovo. The hatched chickens from RFO and Control groups were distributed into two separate houses, each housing 24,500 chickens. Each chicken house was considered an experimental unit and each chicken house represented a replicate for performance data. The chicken houses were located and managed on the same farm (Drobex-Agro Sp. z o.o., Solec Kujawski, Poland), within 20 min driving distance to the laboratory facilities. The treatments received approval of the Polish Local Ethical Commission (No. 22/2012. 21.06.2012) The animals were reared until 42 days of age with continuous ventilation, litter care and starting stocking density of ca. 22 birds/m^2^ (46.39 kg/m^2^), as to obtain a typical production condition, in accordance with the European Union (EU) directive 86/609/EEC. The temperature during onset of farm was 33 °C and was gradually decreased to 20 °C on day 42. The lighting program was initially set for 23L:1D and changed to 18L:6D after the first week. The chickens were offered the standard grower feed mixture with the same chemical composition as provided in Maiorano et al., 2017 [14]. The feed and water we provided ad libitum. The performance data was collected from the Production Process Registration System (SRPP-MK), directly from the slaughterhouse (Drobex-Agro). Livestock body weight (BW), average BW, final stocking density (kg of live weight/m^2^), mortality, feed intake (FI) and feed conversion ratio (*FCR*), were calculated on a weekly basis. The European Broiler Index (*EBI*) was calculated per each chicken house as follows, where viability [%] = living chickens at the end of rearing [%] (1): (1)“EBI=viability” [“%”]“×” “Average Daily Gain” [(“g”/“chicken”)/“day”]/“FCR” “×10” 

### 2.3. Gut Morphology and Microstructure

#### 2.3.1. Histomorphological Samples 

The samples for histological analysis (*n* = 60) were collected at day 21 and day 42, directly from the slaughtered birds (15 chickens per treatment). Samples for histomorphometrical analyses (approx. 2 cm) were taken from the midpoint of the duodenum. In addition, the duodenum, jejunum, ileum, cecum and colon were collected to measure their length and weight.

#### 2.3.2. Histological Preparations 

The duodenum specimens were rinsed with 0.9% NaCl and fixed with the 4% CaCO_3_ buffered formalin solution. The fixed samples were dehydrated, cleared and infiltrated with paraffin in a tissue processor (Thermo Scientific, Waltham, MA, USA), and embedded in paraffin blocks using the embedding equipment (Medite, Burgdorf, Germany). The semi-serial sections of 10 μm thickness were produced from the paraffin blocks using a rotary Thermo Shandon microtome (Thermo Scientific, Waltham, MA, USA). The sections were sequentially mounted on the glass slides coated with albumin/glycerol solution.

#### 2.3.3. Staining Methods and Histomorphometry

Prior staining, the preparations were dewaxed and rehydrated, followed by periodic acid-Schiff (PAS, Sigma-Aldrich, Poznan, Poland) staining to perform morphometric analyses.

The villus height was measured from 10 randomly chosen villi of the cross-sectional area (CSA). This parameter was measured from the tip to the base of the villus at the opening of the crypt. The villus width was measured at its midpoint. The villus surface area was calculated using the formula cited by Sakamoto et al., 2000 [17]: (2)(2π)×(VW/2)×(VH)
where *VW* = villus width, and *VH* = villus height. The intestinal crypt depth was defined as the depth of the invagination between the adjacent villi; the measures were taken from between 10 adjacent villi [18]. The image analysis of the duodenum microstructure was performed using Carl Zeiss microscope (Jena, Germany) and a computer-based image analysis system MultiScan 18.03 (Computer Scanning Systems II, Warsaw, Poland).

### 2.4. Evaluation of the Oxidative Stability of The Meat 

The *Pectoralis superficialis* muscle samples were collected at slaughter (42 day) and analyzed after storage at 4 °C for 0, 2, 4 and 6 days. A hot carcass weight was recorded and carcass yield percentage was calculated. Lipid oxidation was determined by applying a thiobarbituric acid reactive substances (TBARS) assay as described by Vyncke, 1970 [19]. Muscle samples (5 g) were homogenized with the extraction solution (7.5% TCA—trichloroacetic acid, 0.1% Propyl gallate (PG), 0.1% disodium edetate (EDTA)), followed by filtration and mixing with 0.02 M solution of 2-thiobarbutiric acid. The absorbance of the solution was determined spectrophotometrically (UV 8500, Techomp, Japan) at two different wavelengths: 600 nm and 532 nm. The malondialdehyde (MDA) concentration was determined from the standard curve of tetraetoxypropane (TEP). The results were expressed as milligram of MDA per kilogram of tissue. The MDA concentration increases due to degradation of unstable lipid peroxides.

### 2.5. Blood Biochemical Analysis

Blood samples (*n* = 72) were collected from the caudal vein from 12 birds per group, at days 1, 21 and 42 of rearing. The samples were centrifuged after thrombus formation to separate the serum. The protein levels (total protein, albumin, globulin), bilirubin pigment, triglycerides, total cholesterol, lipoproteins, aspartate aminotransferase, alanine aminotransferase and Ca, Fe levels in the serum were measured using a MINDRAY BS-120 biochemical analyser and the reference reagents (STAMAR, Dabrowa Gornicza, Poland).

### 2.6. Coccidia and Bacteria Quantification

#### 2.6.1. Microbiological Sampling

For coccidia counts, the fecal samples were collected weekly from the experimental chicken houses with McMaster chamber procedure according to the Raynaud flotation technique, starting from day 21. The highest susceptibility to coccidia infections is typically observed at the age of 3–6 weeks. The samples for bacteriological quantification: *Bifidobacteria* spp., *Lactobacilli* spp., *Clostridium perfringens, Enterobacteriaceae* and *Campylobacter* (5 replicates per each in ovo group), were collected randomly by two persons on days 21 and 42 of rearing.

#### 2.6.2. Microbiological Analyses

The microbiological examinations were carried out in accordance with ISO 7218:2007/AMD 1:2013 methodology (Microbiology of food and animal feeding stuffs—General requirements and guidance for microbiological examinations). The sampled feces were weighed and diluted. The initial dilutions for analysis were prepared as 1:10 dilution of fecal sample with buffered peptone water (ARGENTA Cat. No. BM0082). The subsequent 10-fold dilutions were prepared using 9 mL of buffered peptone water plus 1 mL of initial dilution. This operation was repeated until reaching a 10^−9^ dilution.

The enumeration of *Enterobacteriaceae* was conducted in accordance with the ISO 21528-2:2004 methodology, applicable to environmental samples in the area of primary production. The suspension culture (1 mL) was started from the prepared 10^−9^ dilutions in sterile Petri dishes. The vessels were filled with VRBG medium (agar with violet, red, bile salts and glucose, ARGENTA Cat. No. BM07508) and the inoculated material was thoroughly mixed with the culture medium. After a complete solidification, the plates were again quenched with VRGB medium and incubated at 37 °C for 24 ± 2 h. For counting, five the most characteristic colonies were selected from each dish and screened on TSA medium (ARGENTA Cat. No. PO512A). After incubation at 37 °C for 24 ± 2 h, the oxidase test (Merck Cat. No. 1.133300.0001) was conducted followed by the selective screening on agar medium with glucose (BTL cat. PWP0081). The fermentation of glucose was observed in all the tubes. All the colonies grown on the VRBG substrate were defined oxidase-negative, while glucose-positive bacteria were classified as belonging to the *Enterobacteriaceae* family.

The enumeration of *Bifidobacterium* spp. was based on a spread plate method with MRS agar (EN 15785:2009). The suspensions of 0.1 milliliter were aspirated from the prepared serial dilutions and inoculated to the AMRS (DeMan, Rogosa and Sharpe agar) substrate with reduced pH 5.7 (ARGENTA Cat. No. BM08908), followed by distribution in a drop-wise manner over the surface of the culture plate. The culture plates were placed in an anaerobic atmosphere (ARGENTA Cat. No. AN0020D) and incubated at 37 °C for 36–48 h. For counting, five out of the most characteristic colonies were picked up from each plate and subjected to Gram staining. *Bifidobacterium* spp. colonies were defined based on the morphological assessment of the test-derived candidate colonies.

The enumeration of *Lactobacillus* spp. was performed in accordance with the EN 15785: 2009 protocol, as above. The culture plates were placed in an anaerobic atmosphere and incubated at 37 °C for 36–72 h. After incubation the colonies were counted at the appropriate dilutions and confirmatory tests were carried out. Five out of the most characteristic colonies were picked up from each plate and subjected to Gram staining and the catalase test. The catalase-negative bacteria, morphologically corresponding to Gram-positive non-sporous bacilli were considered as belonging to the *Lactobacillus* spp. family. 

The counts of *Campylobacter* spp. were undertaken in compliance with the plating method of the PKN-ISO/TS 10272-2: 2008 protocol. Suspension inocula of 0.1 milliliter were aspirated from the prepared serial dilutions and inoculated to charcoal cefoperazone deoxycholate-modified agar substrate (mCDAA, ARGENTA Cat. No. PO5091A). The inocula were distributed in a drop-wise manner over the surface of the culture plates. The culture plates were incubated in a microaerophilic atmosphere (ARGENTA Cat. No. CN0020C) at 41.5 °C for 48 h. After incubation, five out of the most characteristic colonies were picked up from each plate and subjected to oxidase detection, motility testing as well as to evaluation for the ability to grow at 25 °C (microaerophilic atmosphere) and 41.5 °C (oxygen atmosphere). The colonies showing the specific morphological features with characteristic motility, oxidase-positive and non proliferating at 25 °C (microaerophilic condition) and 41.5 °C (aerobic condition) were considered as belonging to *Campylobacter* spp.

The enumeration of *Clostridium perfringens* was performed in compliance with the PN-EN ISO 7937:2005 protocol based on a plating method. Suspension inocula of 0.1 mililiter were aspirated from the prepared serial dilutions, each inoculated to TSC medium supplemented with cycloserine (TSC ARGENTA Cat. No. BO0634M). The culture plates were placed in a microaerophilic atmosphere (ARGENTA Cat. No. CN0020C) and incubated at 37.0 °C for 20 ± 2 h in anaerobic atmosphere (ARGENTA Cat. No. AN0020D). After incubation, five out of the most characteristic colonies were picked up from each plate and the biochemical assessment using the commercial ANA II tests (ARGENTA Cat. No. R8311002) was conducted.

### 2.7. Statistical Analyses

The data were analyzed by one-way analysis of variance (ANOVA), in which the RFO prebiotic was the main factor. The results were verified by conducting statistical analysis in accordance with the methods of statistical inference [20]. The basic statistical descriptors included mean values $\bar{x}$ and standard deviation (±SD). For the other statistical analyses, the values were log-transformed log(x+1) [21]. Normality of the distribution was tested with the Kolmogorov–Smirnov test, while the homogeneity of variance in the samples was assessed with Levene’s test. ANOVA was used to find significant differences between the means and in the case of significant differences Tukey post-hoc test was employed. The level of significance for all the statistical tests was accepted at α = 0.05. For the performance parameters (e.g., mortality, *FCR*) each chicken house was considered the experimental unit and the significance of differences between the means was tested. For other analyses, each individual bird was considered the experimental unit. The statistics were computed using the STATISTICA 13.1 software (Dell, Round Rock, TX, USA, 2018) and MS Excel 2016 software (Microsoft, Redmond, WA, USA).

## 3. Results

### 3.1. Performance Parameters (Mortality, Body Weight (BW), Feed Conversion Ratio (*FCR*) and Feed Intake (FI)) after In Ovo Delivery of Raffinose Family Oligosaccharides (RFO) Prebiotic 

The performance results: hatchability and mortality after the onset on farm (Table 1) and productive performance (Table 2) were not affected by the in ovo RFO prebiotic treatment in this study. The results are considered typical for a Ross308 breed in a normal intensive-rearing scheme (according to Ross308 performance objectives, [22]). Reduction in mortality by 0.30% was observed at day 7 after the onset of hatchlings on farm, and reduction in mortality by 0.40% across the rearing period was observed in the RFO group, but the differences were not significant compared to the Control group. The productivity indices BW, FI and *FCR* did not differ between the RFO and Control groups.

### 3.2. Oxidative Stability of Meat

The analysis of lipid oxidation in meat demonstrated an increased level of peroxidation (*p* = 0.044) of polyunsaturated fatty acids in the RFO group at the beginning of storage (0.07 mg MDA/kg of meat), compared to the Control (0.02 mg MDA/kg of meat) (Figure 1a). The malondialdehyde concentration increased over time due to the degradation of unstable lipid peroxides. The progression of oxidation by day 6 (144 h) became normalized and did not differ in the RFO group compared to the Control group (*p* = 0.433). In general, within the Control group the TBARS values were similar between 0 and 48 h of storage, but the oxidation increased between 0 and 96 h (*p* = 0.001); a further significant progression of oxidation was observed at 144 h in comparison with 0 (*p* = 0.000) and 48 h (*p* = 0.003) (Figure 1b). 

The TBARS values in the RFO group were similar across 0, 48 and 96 h of storage and also between 96 and 144 h. Similar to the Control, a further progression of oxidation was observed at 144 h and it happened to be more evident compared with progression between 0 (*p* = 0.000) and 48 h (*p* = 0.001) (Figure 1b). 

### 3.3. Gut Morphometry and Microstructure

The results of histomorphology of the chicken duodenum, villus height, width and surface area, and the depth of crypts are presented in the Table 3 as mean values. The analyses showed a significant increase in villus height in the RFO group at day 42 (*p* = 0.014) (Figure 2). Also, the villus surface area significantly increased in the RFO group at day 21 (*p* < 0.001) and showed a numerical trend for increase on day 42, compared to the Control group. The crypt depth significantly increased in the RFO group at day 21 (*p* < 0.001) and reached a similar characteristic to the Control group by the end of rearing. 

### 3.4. Biochemical Profile in Blood 

Biochemical blood parameters are presented in Table 4 and Table 5. The levels of proteins, lipids and hepatic indices were measured across the rearing period on days 1, 21 and 42. 

At day 1 of rearing, the level of albumin was elevated after the RFO prebiotic treatment compared to the control group. As a general trend in both groups, a gradual increase in the protein, albumin and globulin levels was observed across the rearing period. Those differences were more pronounced in the Control group (Table 5). At day 1 of rearing, the level of albumin was significantly elevated after the RFO prebiotic treatment compared to the Control group (*p* = 0.033). At day 42, the levels of serum proteins were significantly lower in the RFO group compared to the Control (*p* = 0.002). As a general trend for the lipid profile, a gradual decrease of cholesterol (CHL), high-density lipoprotein (HDL) and low-density lipoprotein (LDL) was observed in both groups from 1 to 42 day (Table 5). For most of the lipid parameters, the values tended to be numerically lower in the RFO group compared to the Control, except for HDL and LDL (day 1) as well as for CHL and HDL (day 42). The level of a beneficial cholesterol fraction (HDL) was significantly higher in the RFO treated chickens compared to the Control group at the end of rearing (*p* = 0.047). The positive changes observed in biochemical profile following the in ovo RFO treatment are additionally indicated in Table 4.

As shown in Table 5, the level of bilirubin was found to be lower in the RFO group at days 1 (*p* = 0.018) and 42 (*p* = 0.006), and the other hepatic indicators aspartate aminotransferase (AST) (*p* = 0.002) and alanine aminotransferase (ALT) (*p* = 0.001) were significantly lower in the RFO-injected chickens at the end of rearing (at 42 days).

At day 42 the serum Ca significantly increased (*p* = 0.032) in the chickens that were in ovo injected with RFO.

### 3.5. Bacteriological Counts 

No increase in colonization with *Lactobacilli* and *Bifidobacteria* was observed after in ovo treatment with the RFO prebiotic (Table 6). For pathogenic species, both groups had negligible *Campylobacter* levels on day 21 of growth. The *Campylobacter* level was not reduced in the RFO group at the end of the growing period. A highly beneficial (Table 4) reduction in *Clostridium perfringens* down to negligible levels (<1, 0 × 10^1^ cfu/g) (*p* = 0.026), was observed in the RFO group across the rearing period (Table 6).

### 3.6. Oocyst Counts

Coccidia oocysts were significantly reduced at days 35, 38 and 42 by RFO prebiotic injected in ovo (Table 7). 

## 4. Discussion

### 4.1. The Performance Was Not Affected by In Ovo Injected RFO

The in ovo injection of RFO prebiotic at day 12 of egg incubation did not affect the overall performance of a large cohort (±25,000) of the hatched chickens. Commercial broilers have been genetically selected for the optimal body weight, feed intake, *FCR* (1.611 for Ross 308 at 6 weeks of age) and other performance parameters. It is rather unexpected to observe significant changes in production parameters after the treatments with natural substances, unless certain clinical symptoms are induced with pathogenic or thermal stress factors. According to Ferket and Gernat [23], it is usually difficult to see the effects of dietary supplements in unstressed broilers. 

Berrocoso et al. [24] delivered increasing doses of raffinose family oligosaccharides in ovo to Cobb500 breeder eggs and did not observe influence on growth performance or slaughter yield of the broilers. The origin of RFO was not given. The prebiotic improved intestine structure of the ileum and immune response indicators. Berrocoso et al. [24] concluded that the injection of RFO during embryogenesis may be a worthwhile strategy for early programming of gut health.

Also, here the injection of RFO did not significantly affect the body weight of broiler chickens, but it improved the microstructure of the duodenal mucosa. Similarly, Pilarski et al. [25] reported that the in ovo injection of RFO did not have a significant effect on body weight of broiler chickens at 42 days of age.

Lipid oxidation is a relevant determinant of deteriorating quality for meat and meat products, as it may cause lipid rancidity [26]. The higher level of peroxidation found in the breast muscle from the RFO prebiotic group is in agreement with a previous study [14], which reported the same trend in the meat of chickens injected in ovo with different commercial prebiotics (DiNovo and Bi2tos). It was suggested that the meat of chickens treated with prebiotics that had a bigger/heavier breast muscle, showed a higher susceptibility to oxidation. However, the highest TBARS values found in this study still remain lower than the accepted limits of TBARS for rancidity (1.0 mg/kg) [27].

### 4.2. Structural Parameters of Gut Health

Our results are mostly in line with the findings of Berrocoso et al. [24]. We found several beneficial effects in the adult birds related to gut health (Table 4), after a single in ovo injection of RFO prebiotic on day 12 of egg incubation: 1/ the villi surface in the duodenum increased at day 21, raising the potential for nutrients absorption from enterocytes, 2/ the weight and length of the cecum, duodenum, jejunum, ileum and colon showed a tendency to decrease by day 42. We explain this by a well-balanced load of the intestine microbiota (not overloaded with bacteria) and a well-developed gastrointestinal tract (GIT), providing that the performance of the physiological biochemistry of the birds was optimal. Earlier, several authors studied the effect of prebiotic oligosaccharides on morphology of chicken GIT [28,29]. They suggested that the birds with well-developed gizzards may not need to modify the length or weight of the small intestine, provided that the gizzard grinds the feed elements small enough for efficient absorption of nutrients. It could be further suggested that the increase in gross morphology of the intestine may be considered an adaptation to dietary challenges in birds that might have failed to develop a highly functional GIT in the critical peri-hatch period, maybe owing to the stresses of the hatching window. Still, in many studies, the increase in weight of the small intestine, especially the cecum, after enzymatic, prebiotic and probiotic treatments, is considered beneficial and is attributed to the increased mass of colonizing probiotic bacteria. However, Gao et al. [30] explains that the extensive bacteria growth is not a normal balance state for the intestine and the increase of intestine mass may result from bacterial infections. Also, the increased density of the diets resulted in intestinal enlargement, explained as the adaptive response to cope with an increased nutrient concentration in the diet [31].

For duodenum histomorphometry, we found decreased length and mass of the duodenum at day 42. The chicken organism has compensated this with the significantly higher villi at day 42 (1586.8 µm in the RFO group vs. 1437.2 µm in the Control). According to Rebole et al. [32] the increase of villi height is accompanied by a tendency to decrease in intestinal crypt depth, which contributes to better utilization of nutrients. 

We observed deeper crypt depth at day 21 in the RFO-treated group. According to Gao et al. [30] the deeper crypts indicate the faster turnover of the intestinal mucosa layer for villus renewal, being a host’s response after injury, e.g., atrophy caused by inflammation from pathogens. Indeed, we assume that the mucosa layer could be stressed by elevated coccidia counts that we found at day 21 in this study (Table 6). On the following days, we found coccidia efficiently reduced in the RFO group followed by normalization of the gut microstructure in the duodenum by day 42.

Pacifici et al. [33] used the in ovo model to study the effect of intra-amniotic raffinose administration (at day 17.5 of egg incubation), on the Fe status in vivo, the brush border membrane and reduction of pathogenic bacteria like *Clostridia* and *E.coli*. In their study, the villi surface was increased at a day of hatch, after the in ovo injection of the raffinose. Also, the cecum weight increased, explained by increased number of colonizing bacteria populations (*Lactobacilli, Bifidobacteria*). 

### 4.3. The Improved Biochemical Profile in Blood

In this study, we confirm the positive effect of a single in ovo probiotic treatment, on shaping the biochemical profile in growing broiler (Table 5). At day 1 of rearing, the level of albumin was elevated after the in ovo RFO prebiotic treatment compared to the Control group, which could point to the intensified synthesis in liver, to provide the body with building material. As explained elsewhere, the alterations in biochemistry in young broilers are especially pronounced during adaptation to a new feeding regime [34]. The increase of serum proteins and albumin not only reflects the status of protein synthesis, carbohydrate, and lipid metabolism [35], but according to Alonge et al., it also indicates the status of resistance and immune response in broilers [35]. In this study, the gradual increase in protein levels in the fattening period can be considered normal and related to the intensive growth of broilers, with similar results as in the other studies conducted on the Ross38 breed [34,35,36,37]. However, we observed a lower protein level in the serum of RFO-injected birds compared to the Control group at day 42 of rearing. This might result from a good metabolic turnover in the RFO-treated birds, which allowed for better utilization of the nutrients and a lower requirement for protein synthesis at the end of fattening, thus helping to maintain the metabolic balance during the period of rapid growth [34]. It is the more so because the other biochemical indicators show for optimal homeostasis in the RFO-injected birds: a beneficial lipid profile and good hepatic indices (also discussed further), as well as the good performance parameters. 

The RFO extracted from lupin improved the lipid profile in blood by an increased proportion of HDL (67.7 mg/dL, Table 4) and decreased levels of unfavorable low density lipoproteins, by day 42. A similar effect was reported earlier for the RFO extracted from pea and injected in ovo to Hubbard breeder eggs [25]. The possible mechanisms for prebiotic action can be explained by its role in the reduction of synthesis and absorption of cholesterol and LDL; either by inhibition of hepatic synthesis through hydrolytic deconjugation of cholesterol from bile salts or by inhibition of 3HMG-CoA—an intermediate in the mevalonate pathway leading to synthesis of cholesterol from acetylo-CoA [38]. 

We found, that the in ovo delivered RFO reduced the levels of bilirubin, ALT and AST at the end of rearing. The low levels of these biochemical indices reflect a good condition of liver and muscles [36,39]. Such a hepatoprotective effect of the RFO prebiotics was also confirmed in other studies. Zhang et al. [40] also showed an antioxidant and hepatoprotective effect of the RFO extracted from *Rehmannia glutinosa* Libosch herb in the challenged (CCl_4_-intoxicated) mice. No effect was found in turkeys fed a yellow lupin seed meal [41].

Here, the RFO increased calcium level in the serum at day 42. The enhanced calcium absorption from intestine tract caused by prebiotic supplementation was confirmed by Hatab et al. [42] and Scholz-Ahrens et al. [43]. Summarizing, the biochemical results confirmed a good health condition of the birds reared in this experiment, with the positive effects of a single, in ovo dose of the RFO prebiotic, pronounced in the end of rearing.

### 4.4. The Mitigation of Natural Environmental Pathogenic Infections and Opportunistic Pathogens

The effects of reducing parasitic oocysts by the prebiotics are likely to provide benefits to the health of the flock [10]. The oocyst shed in broilers droppings. At day 42, a significant, 89% reduction in oocysts was observed after the lupin RFO treatment in ovo. This is in line with the study of Angwech et al. [10] where another prebiotic, the galactooligosaccharides injected in ovo on day 12 of egg incubation, reduced the shedding of oocysts in excreta of Kurolier (slow growing) chickens after 6 weeks of rearing. 

The optimal time to achieve reduction in oocyst numbers is in growing chickens and young adults. Several studies have shown that litter oocyst counts in commercial broiler houses may exceed 10^5^ oocysts/g at 4 to 5 weeks of age [44]. It is generally accepted that the immunity of chickens is acquired not earlier than at 7 weeks of age and, thus, the period of 3–6 weeks of age is critical for the chickens to fail to resist to clinical infections. In order to control coccidiosis, probiotics and prebiotics are considered alternative sustainable strategies apart from preventive treatments with live attenuated vaccines. The *Eimeria* sporozoites that release from sporocytes enter the intestinal cells and multiply there. The assumed mechanism of prebiotic action is an indirect inhibition of enterocyte penetration through immune stimulation and blocking binding the infectious agent to a mucosal surface of the intestine. However, variable results were obtained on a mixed infectious model with *Eimeria acervulina, E. tenella and E. maxima* using mannano-oligosasccharides [45] A significant reduction of lesions’ score and oocyts shedding by the alphamune prebiotic (beta-glucans and oligosaccharides) alone and combined with a ionophore- based coccidiostat [46]. 

Here, a single dose in ovo of the RFO prebiotic showed 2.5 log reduction in Clostridium perfringens at day 21 of growth. The reduction of Clostridium by raffinose in the hatched chickens was also reported by Paciffi et al. [33] after the manual administration of 1 mL soluble powder raffinose to the amniotic fluid, on day 17.5 of egg incubation. 

According to Porter et al. [47] *C. perfringens* poses the main health problem associated with removing the antibiotics from feed. The majority of *Clostridia* types are commensal, but potentially opportunistic gut microbiota that are abundant in the gut of healthy animals. The pathological clinical symptoms result from the production of toxins and associated by co-existing coccidiosis or immunosuppression with environmental stressors. Until modification of the sporulation medium, the regulation of *C. perfringens* activity in vitro was problematic. By replacing starch with raffinose, Labbe and Ray [48] obtained the accelerated sporulation of *C. perfringens* in Duncan and Strong medium in vitro and the increased toxin production. In vivo (in ovo), the addition of RFO prebiotic seem to beneficially shift the bacteria colonization in broiler gut to significantly reduce the number of *C. perfringens*. Apart from our result, it is hard to find a confirmation in literature for the raffinose action against *Clostridia*. Zdunczyk et al. [49] showed a decrease of *E.coli, Bacteroides, Prevotella* and *Porphyromonas* in laying hens fed 20% inclusion of lupin seeds to the diet. Also Davani-Davari et al. [50] in their latest review on prebiotics report that the effect of RFO on gut microbiota has still not been elucidated enough and further research is required. In a clinical study, the administration of raffinose to healthy volunteers resulted in a significant increase in faecal *Bifidobacteria* and decrease of *Bacteroides* and *Clostridia* [51]. In another human study, the presence of *Clostridium histolyticum* and *Clostridum lituseburense* groups was reduced in the chickpea–raffinose containing diet [52]. The mechanism of action of raffinose may be assumed to act similar to the other reported non-starch oligosaccharides: they 1/ inhibit growth of pathogens through increased fermentation by *Lactobacillus* and *Bifidobacteria* and production of short chain fatty acids (SCFA), which competitively reduces growth of *C. perfringens, E. Coli* and *Salmonella*, increases villi height, and MUC gene expression, 2/ modulate immune response through increase of IgA, IgM, IgG and decrease IL-6 and IFN-γ [6]. Our previous studies allow suggesting that in ovo injected RFO prebiotic may influence the microbiota composition and pose immunomodulatory effects later in a chicken’s life. Previously, a single injection in ovo of a synbiotic based on RFO differentially modulated immune-related genes expression; IL-12 gene expression was silenced after 3 and 6 weeks of rearing in Cobb broilers [53]. In ovo injection of RFO prebiotic alone or in a synbiotic combination with *Lactobacillus plantarum* had an effect on the immune system development after 3 and 6 weeks of rearing. The bursa development was faster, the bursa to spleen index was higher, and an increased number of developing T-cell progenitors in the thymus was observed [54]. A range of other life-long phenotypic effects after in ovo delivery of RFO compound is also listed in Table 1 in Siwek et al. [7]. 

Based solely on the research committed to in ovo injection of RFO, the following mechanism of action of RFO can be suggested:

1/competitive reduction of access to colonization sites for potential opportunistic pathogens by promoting the proliferation of *Bifidobacteria* at hatching [15,25];

2/ improving the ileum mucosa morphology and immunity in the small intestine [8,24];

3/stimulation of the immune system by accelerated development of lymphatic organs and impact on cytokines genes expression in spleen and cecal tonsils [53,54].

### 4.5. ‘The Earlier The Better Effect’ of In Ovo Prebiotic Treatments

At hatching, the microbiome composition is of low complexity and susceptible to dietary and management manipulations of colonizing the unsettled niches in the gut. However, the microbiota composition is highly dependent on the age of the individual, and it was proved that the effect of the host age is stronger than the post-hatch bacteria treatments (probiotic or live attenuated Salmonella vaccine) provided after hatching [55]. We support the hypotheses that the most promising strategies to modulate gut health effectively are based on in ovo applications [24,56]. Different approaches have been discussed elsewhere [7,57]. Still, the extra labour effort required to inject the eggs at day 12 of incubation limit the implementation of this injection time-point to a common practice. The factors are: an extra effort to remove the egg trays from the incubator and a 2-step injection procedure: injection itself, and sealing the injection hole afterwards (Video S1). Instead, at 17.5–19.2 days of egg incubation, the probiotic or a synbiotic supplementation can be potentially combined with an automatic in ovo vaccination. Further industrial research is required to prove the efficacy of such a combined treatment. Technically, there is a high probability of a missed injection site of prebiotics and probiotics—not to the amnion but to the muscles or allantoic sac. In our opinion, the further scientific rationales for advancement and adaptability of in ovo strategy to modulate gut health and resistance at the earliest time-points (12 day) shall be provided, comprising the industrial technological adaptation.

## 5. Conclusions

This investigation provided further proof that a single in ovo dose of a natural bioactive compound delivered on day 12 of egg incubation beneficially modulated systems of the chicken organism and did not affect the optimal performance. The holistic condition was reflected in the improved blood lipid profile, gut structure and better resistance against the natural opportunistic pathogens in the rearing environment. The raffinose family oligosaccharides sourced from the seeds of the local legumes can be considered a promising prebiotic for broiler chickens, provided that the extraction technology is scaled-up for supplementation of large quantities in feed. Optionally, the in ovo injected dose is several hundred times lower and provides lifelong effects. 

## Figures and Tables

**Figure 1 animals-10-00592-f001:**
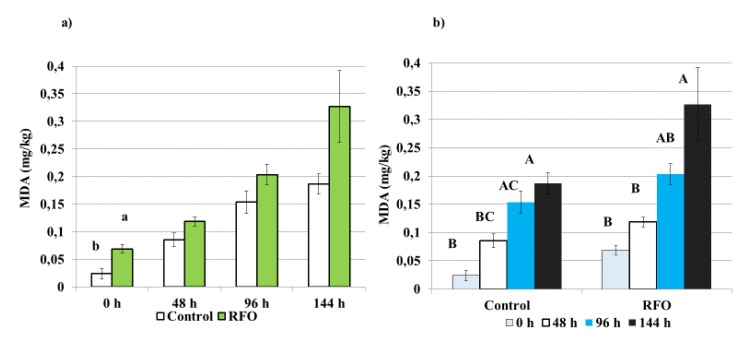
Effect of raffinose family oligosaccharides (RFO) prebiotic treatment on thiobarbituric acid-reactive (TBA) substances values (mg malondialdehyde (MDA)/kg of meat) in breast muscle, *Pectoralis superficialis* (**a**) and lipid oxidation progression in breast muscle within each treatment (**b**), during the storage at 4 °C. TBA values were compared between RFO prebiotic and the Control treatments on days 0 (0 h), 2 (48 h), 4 (96 h) and 6 (144 h) of meat storage. The assessment involved 2 separate houses, one for prebiotic and one for the Control. The replicate was TBA-reactive substances values per randomly selected bird. There were *n* = 15 birds selected for RFO and Control group. For each bird, there were 3 samples (repetitions) taken for each storage time point and each sample was analyzed in duplicate. The charts represent means ± standard error (SE) bars. A, B, C: *p* < 0.001; a,b: *p* < 0.05.

**Figure 2 animals-10-00592-f002:**
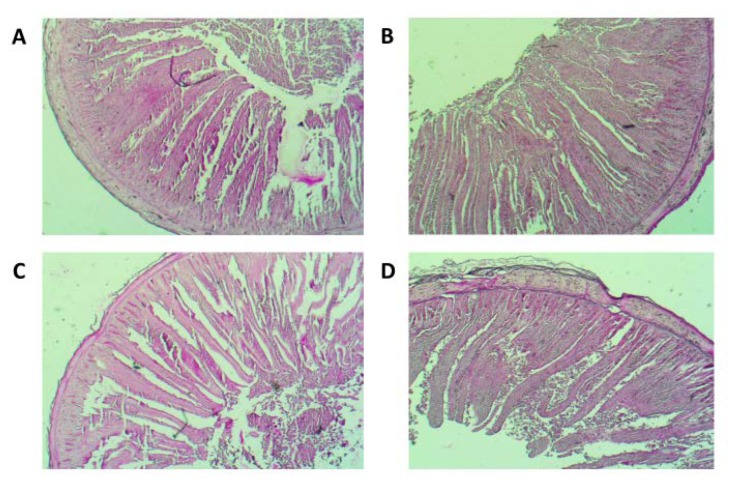
Effect of the in ovo RFO prebiotic treatment on the histomorphology of duodenum in broiler chicken. (**A**) Photomicrograph (light microscope) of the duodenum in the RFO group on day 21. Periodic acid-Schiff (PAS) reaction, magn. ×32; (**B**) photomicrograph (light microscope) of the duodenum in the RFO group on day 42. PAS reaction, magn. ×32; (**C**) photomicrograph (light microscope) of the duodenum in the Control group on day 21. PAS reaction, magn. ×32; (**D**) photomicrograph (light microscope) of the duodenum in the Control group on day 42. PAS reaction, magn. ×32.

**Table 1 animals-10-00592-t001:** Mortality of chickens and housing condition.

Group	No.Chick/ House at Day 1	No.Chick/ m^2^ at Day 1	Final No.Chick/ House at Day 42	Hatchability (%)	Mortality at Day 7 (%)	Total Mortality (%)
RFO	24500	22	22769	84.56	1.78	2.9
Control	24500	22	22705	85.35	2.08	3.3
SE	n/a	n/a	32	0.3950	0.15	0.2
*p*	n/a	n/a	0.2952	0.2958	0.2952	0.2926

RFO—group injected with raffinose family oligosaccharides at day 12 of egg incubation; Control—group injected with physiological saline (0.9% NaCl). Significance: *p* ≤ 0.05.

**Table 2 animals-10-00592-t002:** Productive performance: mean feed intake, feed conversion ratio, body weight and European Broiler Index.

Group	Average BW (kg/Bird)	FI (g/day/Bird)	FCR (kg/kg)	EBI
RFO	2.24	94.20	1.72	322.70
Control	2.30	95.20	1.69	335.50
SE	0.03	0.49	0.01	6.40
*p*	0.2804	0.2952	0.2804	0.2951

RFO—group injected with raffinose family oligosaccharides at day 12 of egg incubation; Control—group injected with physiological saline (0.9% NaCl); BW—body weight; FI—Feed intake; *FCR*—Feed conversion ration; *EBI*—European Broiler Index. Significance: *p* ≤ 0.05.

**Table 3 animals-10-00592-t003:** Effect of the in ovo RFO prebiotic treatment on intestinal structure and duodenum histomorphology of 21- and 42-day-old broiler chickens.

Day of Rearing	Day 21		ANOVA	Day 42		ANOVA
	RFO	Control	*p*	RFO	Control	*p*
Body weight (g)	750.9 ^a^ ± 22.9	744.7 ^a^ ± 9.2	0.338	2249.6 ^a^ ± 535.9	2168.6 ^a^ ± 103.2	0.570
(mm)						
Length of D	24.9 ^a^ ± 2.5	25.2 ^a^ ± 1.4	0.687	28.7 ^b^ ± 2.6	31.9 ^a^ ± 1.7	<0.001
Length of J	61.9 ^a^ ± 3.7	60.1 ^a^ ± 3.4	0.194	65.7 ^b^ ± 4.5	75.8 ^a^ ± 5.0	<0.001
Length of I	61.3 ^a^ ± 5.2	57.8 ^a^ ± 7.9	0.162	67.1 ^b^ ± 3.4	74.6 ^a^ ± 5.0	<0.001
Length of cecum	26.1 ^a^ ± 2.6	25.7 ^a^ ± 2.4	0.649	35.8 ^b^ ± 3.0	38.8 ^a^ ± 2.9	0.009
Length of colon	8.3 ^a^ ± 1.0	8.0 ^a^ ± 0.5	0.275	8.7 ^b^ ± 0,7	10.4 ^a^ ± 0.9	<0.001
Weight of D (g)	6.8 ^a^ ± 1.0	7.3 ^a^ ± 0.6	0.121	12.4 ^b^ ± 1.7	13.9 ^a^ ± 1.4	0.011
Weight of J (g)	12.6 ^a^ ± 1.7	13.1 ^a^ ± 1.3	0.390	23.3 ^b^ ± 3.3	27.5 ^a^ ± 5.0	0.010
Weight of I (g)	10.5 ^a^ ± 1.5	10.5 ^a^ ± 1.1	0.938	19.3 ^a^ ± 2.5	21.2 ^a^ ± 3.2	0.073
Weight of cecum (g)	3.8 ^b^ ± 0.8	4,5 ^a^ ± 1,0	0.029	11.6 ^a^ ± 1.9	11.9 ^a^ ± 2.5	0.743
Weight of colon (g)	2.0 ^a^ ± 0.7	1.8 ^a^ ± 0.4	0.545	3.2 ^b^ ± 0.4	4.0 ^a^ ± 0.5	<0.001
Villus height (µm)	1281.0 ^a^ ± 160.4	1333.9 ^a^ ± 114.6	0.308	1586.8 ^a^ ± 199.7	1437.2 ^b^ ± 94.4	0.014
Villus width(µm)	137.6 ^a^ ± 16.3	106.4 ^b^ ± 7.9	<0.001	111.2 ^a^ ± 7.5	118.8 ^a^ ± 19.8	0.175
Villus surface area (µm)	549,263.3 ^a^ ± 63431.3	442,127.3 ^b^ ± 46013.5	<0.001	549,100.6 ^a^ ± 78401.0	538,983.8 ^a^ ± 107540.2	0.771
Crypt depth (µm)	140.4 ^a^ ± 8.2	124.0 ^b^ ± 11.5	<0.001	166.1 ^a^ ± 28.8	178.4 ^a^ ± 11.2	0.135

RFO—group injected with raffinose family oligosaccharides at day 12 of egg incubation; Control—group injected with physiological saline (0.9% NaCl); ANOVA—Analysis of Variance; D—duodenum, I—ileum, J—jejunum. ^a,b^ mean values with the same letter are not significantly different at *p* ≤ 0.05..

**Table 4 animals-10-00592-t004:** The beneficial effects related to physiological health observed in the chickens after a single in ovo injection with RFO prebiotic.

Days of Rearing	Day 21	Day 35	Day 42
**Gut structure**			
Lenght of D, J, I, C, colon			+
Weight of D, J, colon			+
Weight of cecum	+		
Height of the villi			+
With and surface of the villi	+		
**Biochemical indices**			
HDL			+
ALT			+
AST			+
Ca			+
**Bacteriological counts**			
*Clostridium perfringens*	+		+
**Oocysts counts**		+	+

+ indicates a positive effect. RFO—raffinose family oligosaccharides, D—duodenum, J—jejunum, I—ileum, C—cecum, HDL—high density lipoprotein; ALT—alanine aminotransferase, AST—aspartate aminotransferase, Ca—calcium.

**Table 5 animals-10-00592-t005:** The biochemical profile in growing broiler chickens, injected with RFO prebiotic in ovo, at day 12 of egg incubation.

Day	Group	Protein (g/dL)	Albumins (g/dL)	Globulins (g/dL)	Bilirubin (mg/dL)	Triglycerides (mg/dL)	Cholesterol (mg/dL)	HDL (mg/dL)	LDL (mg/dL)	ALT (u/L)	AST (u/L)	Ca (mg/dL)	Fe (mg/dL)
1	RFO	2.5 ^a^ ± 0.2	1.1 ^a^ ± 0.1	1.3 ^a^ ± 0.1	0.2 ^a^ ± 0.0 *	68.8 ^a^ ± 10.9	474.5 ^a^ ± 21.9	228.0 ^a^ ± 16.6	232.7 ^a^ ± 12.2	17.8 ^a^ ± 1.3	227.8 ^a^ ± 17.5	9.6 ^a^ ± 0.4	103.8 ^a^ ± 59.8
	Control	2.4 ^a^ ± 0.4	1.0 ^b^ ± 0.1	1.3 ^a^ ± 0.2	0.2 ^b^ ± 0.0 *	79.3 ^a^ ± 11.5	484.5 ^a^ ± 50.0	187.7 ^a^ ± 137.0	152.3 ^a^ ± 130.9	20.9 ^a^ ± 5.3	325.6 ^a^ ± 253.1	10.0 ^a^ ± 0.4	90.4 ^a^ ± 54.4
	*p*	0.794	0.033	0.674	0.018	0.112	0.657	0.492	0.163	0.197	0.368	0.087	0.668
21	RFO	2.6 ^a^ ± 0.7	1.3 ^a^ ± 0.5	1.2^a^ ± 0.2	0.2 ^a^ ± 0.0	78.9 ^a^ ± 22.6	121.6 ^a^ ± 15.0	66.7 ^a^ ± 10.3	39.2 ^a^ ± 6.1	14.4 ^a^ ± 1.9	237.5 ^a^ ± 24.8	10.5 ^a^ ± 0.7	92.8 ^a^ ± 60.5
	Control	2.9 ^a^ ± 0.4	1.6 ^a^ ± 0.2	1.3 ^a^ ± 0.2	0.2 ^a^ ± 0.0	87.3 ^a^ ± 24.5	137.9 ^a^ ± 26.3	79.8 ^a^ ± 16.1	40.7 ^a^ ± 9.3	15.5 ^a^ ± 2.5	233.4 ^a^ ± 35.3	10.2 ^a^ ± 0.8	123.3 ^a^ ± 62.8
	*p*	0.152	0.120	0.341	0.689	0.450	0.131	0.058	0.688	0.292	0.780	0.369	0.293
42	RFO	3.1 ^b^ ± 0.2	1.6 ^b^ ± 0.0	1.5 ^b^ ± 0.2	0.1 ^b^ ± 0.0	67.4 ^a^ ± 13.5	119.4 ^a^ ± 14.0	67.7 ^a^ ± 10.8	38.2 ^a^ ± 8.0	10.9 ^b^ ± 1.4	245.1 ^b^ ± 24.1	11.0 ^a^ ± 0.8	166.1 ^a^ ± 84.3
	Control	3.6 ^a^ ± 0.3	1.8 ^a^ ± 0.1	1.8 ^a^ ± 0.2	0.2 ^a^ ± 0.0	79.4 ^a^ ± 34.2	115.3 ^a^ ± 20.6	48.3 ^b^ ± 22.8	39.6 ^a^ ± 31.8	15.1 ^a^ ± 2.4	353.3 ^a^ ± 75.9	10.2 ^b^ ± 0.5	124.8 ^a^ ± 85.7
	*p*	0.002	<0.001	0.026	0.006	0.372	0.647	0.047	0.909	0.001	0.002	0.032	0.347

* The exact mean ± SD for the RFO was 0.163 ± 0.005 and for the Control was 0.18 ± 0.014; RFO—raffinose family oligosaccharides, HDL—cholesterol high density lipoprotein, LDL—cholesterol low density lipoprotein, ALT—alanine aminotransferase, AST—aspartate aminotransferase, Ca—calcium, Fe—ferrum; ^a,b^ mean values with the same letter are not significantly different at *p* ≤ 0.05.

**Table 6 animals-10-00592-t006:** Quantification of major commensal bacteria strains prone to modulation with prebiotics and representing the bacteriological status of chicken intestine.

Day	Bacteria Groups	RFO (cfu/g)	Control (cfu/g)	*p*-Value
21	*Lactobacillus*	4.6 × 10^8^ ± 2.2 × 10^8^	6.4 × 10^8^± 5.4 × 10^8^	0.733
	*Bifidobacterium*	2.6 × 10^8^ ± 2.8 × 10^8^	2.9 × 10^8^ ± 2.2 × 10^8^	0.385
	*Enterobacterium*	8.9 × 10^7^ ± 7.1 × 10^7^	4.5 × 10^8^ ± 4.0 × 10^8^	0.252
	*Clostridium*	<1.0 × 10^1,b^ ± <1.0 × 10^1^	4.7 × 10^^2,a^^ ± 4.8 × 10^2^	0.026
	*Campylobacter*	<1.0 × 10^1^ ± <1.0 × 10^1^	<1.0 × 10^1^ ± <1.0 × 10^1^	−
42	*Lactobacillus*	1.9 × 10^8^ ± 7.1 × 10^7^	1.6 × 10^8^ ± 1.0 × 10^8^	0.434
	*Bifidobacterium*	<1.0 × 10^1^ ± <1.0 × 10^1^	2.0 × 10^8^ ± 4.5 × 10^7^	0.347
	*Enterobacterium*	3.0 × 10^7^ ± 3.3 × 10^7^	4.4 × 10^8^ ± 4.1 × 10^7^	0.767
	*Clostridium*	<1.0 × 10^1,b^ ± <1.0 × 10^1^	2.2 × 10^^3,a^^ ± 2.9 × 10^3^	0.046
	*Campylobacter*	5.2 × 10^6^ ± 2.5 × 10^6^	2.0 × 10^6^ ± 1.8 × 10^6^	0.125

^a,b^ mean values with the same letter are not significantly different at *p* ≤ 0.05.

**Table 7 animals-10-00592-t007:** Quantification of coccidia oocysts present in feces of in ovo injected chickens.

Day	Group	Oocysts Number	*p*-Value
21	RFO	169,600.0 ^a^ ± 7372.9	0.002
	Control	131,333.3 ^b^ ± 4772.1	
35	RFO	416.7 ^B^± 28.9	<0.001
	Control	716.7 ^A^± 28.9	
38	RFO	516.7 ^b^ ± 28.9	0.013
	Control	616.7 ^a^ ± 28.9	
42	RFO	50.0 ^B^ ± 0.0	<0.001
	Control	450.0 ^A^ ± 50.0	

Differences in oocyst counts between the in ovo RFO and Control treatments. Coccidia oocysts counts were compared between treatments on days 21, 35, 38 and 42. The assessment involved two separate houses, one for each of the treatments. A total of *n* = 5 fecal samples were taken per replicate at random locations in the house (horizontal method) and combined to obtain a representative sample. The replicate used for statistical analysis was the number of oocysts in 1g of fecal sample with 3 replicates per treatment. Mean values with the same letter are not significantly different for ^a,b^ at *p* ≤ 0.05, ^A,B^ at *p* ≤ 0.001.

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
