# Peer review of "Injection of Raffinose Family Oligosaccharides at 12 Days of Egg Incubation Modulates the Gut Development and Resistance to Opportunistic Pathogens in Broiler Chickens"

_animals, 2020, doi:10.3390/ani10040592_

Round 1

Reviewer 1 Report

This is an interesting study of determined the effects of the injection of raffinose family oligosaccharides at day 12 of egg incubation modulates the gut development and resistance to opportunistic pathogens in broilers. The current study found that RFO sourced from local legumes can be considered as a promising prebiotic for broiler chickens. However, there are few technical and editorial issues that should be addressed prior to the manuscript being accepted for publication. These are listed below. 

  1. Line 39, the designation of groups 1/ in ovo_Control and 2/ in ovo_RFO need to be simple, such as Control and RFO. Please go though the text and revised all.
  2. Line 57, please clarified why you injected RFO at the day 12 of egg incubation.
  3. Line 108, please clarified experimental design for the birds, including how many replicates for each treatment and how many birds for each replicate?
  4. Line 265, in Figure legend, what’s the replicates/n=? for the data? What’s the P value for the significant differ here? Please checked for all the figures and tables. Meanwhile, too many lines in the Figure 1.
  5. Table 4, color do not need to be showed in the table.
  6. Line 495, in the discussion section, the author should explained why MDA was higher in the RFO group?

Author Response

Responses to Reviewer 1 Comments (in blue)

Dear Reviewer,

Thank You very much for the evaluation of the manuscript. Please find below the detailed answers to the comments as well as the amendments in the revised version which are proposed for Your further consideration:

Point 1.

Line 39, the designation of groups 1/ in ovo_Control and 2/ in ovo_RFO need to be simple, such as Control and RFO. Please go though the text and revised all.

Response 1.

Thank You for the suggestion. The names of the groups are changed in the revised manuscript in accordance with the suggestion.

Point 2.

Line 57, please clarified why you injected RFO at the day 12 of egg incubation.

Response 2.

Day 12 of egg incubation is the first time point to stimulate native microflora solely with prebiotic or synbiotic, in the prenatal period of a broiler chicken. On a contrary to vitamins or vaccines injected at a later stage of egg incubation (17.5-19 day), the prebiotic or a synbiotic compound must be delivered to the digestive tract of the embryo. On day 12 of egg incubation, the prebiotic compound is deposited in the air chamber and penetrates to a developing gut within several hours, through a vascularized allantochorion (Siwek et al., 2018 [7]). The rationale to inject prebiotic on day 12 of egg incubation is also provided in Material and methods: ‘The rationale to deliver prebiotic in ovo precisely on day 12 of egg incubation was based on a series of experiments reported elsewhere (Villaluenga et al., 2004 [15]; Pilarski et al., 2005 [24]).

Point 3.

Line 108, please clarified experimental design for the birds, including how many replicates for each treatment and how many birds for each replicate?

Response 3.

The experimental design followed that published by Sobolewska et al. 2017 [8]. That study was also performed on a large cohort of the animals and the prebiotic versus Control group were compared: ‘A total of 54,000 eggs, containing living embryos were randomly divided into 2 equal groups: a control group and an experimental group, injected with the prebiotic DiNovo (DiNovo Group)’.

In the revised manuscript, the description is amended as follows:

L 106-109,

The hatched chickens from RFO and Control groups were distributed into two separate houses, each housing 24500,00 chickens. Each chicken house was considered an experimental unit and each chicken house represented a replicate for performance data.

Point 4.

Line 265, in Figure legend, what’s the replicates/n=? for the data? What’s the P value for the significant differ here? Please checked for all the figures and tables. Meanwhile, too many lines in the Figure 1.

Response 4.

Thank You for the comment, we explain the replicates and provide the values of significance in the amended manuscript.

The legend for Figure 1 is amended as follows:

TBA values were compared between RFO prebiotic and the Control treatments on days 0 (0h)- 2 (48h)- 4 (96h) and 6 (144h) of meat storage. The assessment involved 2 separate houses, one for prebiotic and one for the Control. The replicate was TBA-reactive substances values per randomly selected bird. There were n=15 birds selected for RFO and Control group. For each bird, there were 3 samples (repetitions) taken for each storage time point and each sample was analyzed in duplicate. The charts represent means ± SE bars. A, B, C: p < 0.001; a,b: p  < 0.05.

Table 3: the missing units [mm] [g] [µm] are provided.

Table 4: the positive effect on HDL is added (was missing previously)

Table 6: the units for bacteria counts [cfu/g] are provided.

Also, the missing titles in the reference list are provided for Lamot et al., 2019 [30] and Scholz-Ahrens et al., 2007 [42]

The other tables were checked for consistency in providing the p values.

The descriptions of results are improved and the values of significance are additionally included in the descriptions of results:

Section 3.2:

L 261-273,

The analysis of lipid oxidation in meat demonstrated an increased level of peroxidation (p = 0.044) of polyunsaturated fatty acids in the RFO group at the beginning of storage (0.07 mg MDA/kg of meat), compared to the Control (0.02 mg MDA/kg of meat) (Figure 1a). The malondialdehyde concentration increased over time due to the degradation of unstable lipid peroxides. The progression of oxidation by day 6 (144h) got normalized and did not differ in the RFO group compared to the Control group (p = 0.433).

In general, within the Control group the TBARS values were similar between 0 and 48 h of storage, but the oxidation increased between 0 and 96 h (p = 0.001); a further significant progression of oxidation was observed at 144 h in comparison with 0 (p = 0.000) and 48 h (p = 0.003) (Figure 1b).

The TBARS values found in the RFO group were similar across 0, 48 and 96 h of storage and also between 96 and 144 h. Similar to the Control, a further progression of oxidation was observed at 144h and it happened to be more evident compared with progression between 0 (p = 0.000) and 48 h (p = 0.001) (Figure 1b).

Section 3.4:

L 313-318,

As a general trend in both groups, a gradual increase in the protein, albumin and globulin levels was observed across the rearing period. Those differences were more pronounced in the Control group (Table 5). At day 1 of rearing, the level of albumin was significantly elevated after the RFO prebiotic treatment compared to the Control group (p=0.033). At day 42, the levels of serum proteins were significantly lower in the RFO group compared to the Control (p=0.002).

L 322-335,

As a general trend for the lipid profile, a gradual decrease of cholesterol (CHL), high-density lipoprotein (HDL) and low-density lipoprotein (LDL) was observed in both groups from 1 to 42 day (Table 5). For most of the lipid parameters, the values tended to be numerically lower in the RFO group compared to the Control, except for HDL and LDL (day 1) as well as for CHL and HDL (day 42). The level of a beneficial cholesterol fraction (HDL) was significantly higher in the RFO treated chickens compared to the Control group at the end of rearing (p=0.047). The positive changes observed in biochemical profile following the in ovo RFO treatment are additionally indicated in Table 4.

As shown in Table 5, the level of bilirubin was found lower in the RFO group at days 1 (p=0.018) and 42 (p=0.006), and the other hepatic indicators aspartate aminotransferase (AST) (p=0.002) and alanine aminotransferase (ALT) (p=0.001) were significantly lower in the RFO injected chickens at the end of rearing (at 42 days).

At day 42 the serum Ca significantly increased (p= 0.032) in the chickens that were in ovo injected with RFO.

Section 3.5:

L 351-355,

A highly beneficial (Table 4) reduction in Clostridium perfringens as much down as to negligible levels (<1,0x10(1) cfu/g) (p=0.026), was observed in the RFO group across the rearing period (Table 6). 

Point 5.

Table 4, color do not need to be showed in the table.

Response 5.

The color is removed in the revised version of the manuscript.

Point 6.

Line 495, in the discussion section, the author should explained why MDA was higher in the RFO group?

Response 6.

The following paragraph is added to discussion (at the end of Section 4.1):

L 390-396,

Lipid oxidation is a relevant determinant of deteriorating quality for meat and meat products, as it may cause lipid rancidity (Jin et al., 2009 [25]). The higher level of peroxidation found in the breast muscle from the RFO prebiotic group is in agreement with a previous study (Maiorano et al., 2017 [14]), which reported the same trend in the meat of chickens injected in ovo with different commercial prebiotics (DiNovo and Bi2tos). It was suggested that the meat of chickens treated with prebiotics that had a bigger/heavier breast muscle, showed a higher susceptibility to oxidation. However, the highest TBARS values found in this study still remain lower than the accepted limits of TBARS for rancidity (1.0 mg/kg) (Rahman et al., 2015 [26]).

The two following references are added:

[25] Jin, S. K.; Kim, I. S.; Choi, Y. J.; Kim, B. G.; Hur, S. J. The development of imitation crab sticks containing chicken breast surimi. LWT-Food Sci. Technol. 2009, 42, 150-156.

[26] Rahman, M. H.; Hossain, M. M.; Rahman, S. M. E.; Amin, M. R.; Oh, D. H. Evaluation of physicochemical deterioration and lipid oxidation of beef muscle affected by freeze-thaw cycles. Korean J. Food Sci. An. 2015, 35, ( 6), 772-782.

Reviewer 2 Report

General Comments -

The manuscript "Injection of raffinose family oligosaccharides at 12 2 day of egg incubation modulates the gut 3 development and resistance to opportunistic 4 pathogens in broiler chickens" by K. Stadnicka, J. Bogucka, M. Stanek, R. Graczyk, K. Krajewski, G. Maiorano, and M. Bednarczyk describes the effects of a single-dose prebiotic treatment given to eggs on the various measures in chicken. Overall, the manuscript is well presented and experiments conducted are relevant and provide interesting results.

The Discussion section as presented often is limited to presenting summaries of previous studies and lacks a clear and coherent discussion on the interpretation of the results. For example, section 4.1 presents 3 summaries of three previous studies but does not attempt to make a clear connection to the results of these studies. Another issue is there was no data in Table 4.

Table 4 does not have any data.

L63:  Define FCR at first occurrence.

L144:  remove space in "cry  pt"

L237:  Correct spelling "Tuckey" to "Tukey"

L302:  The sentence "At day 42, the levels ........  group." is unclear. Please clarify if the protein levels in RFO group were decreased compared to control or some other point of reference.

L306 -310:  Please revise the paragraph. The sentences, as written, are conflicting and their general meaning is unclear.

Section 3.4: Was statistical analysis was conducted on blood parameters presented in Table 4 (My version does not have any data in Table 4) and Section 3.4. Table 5 presents statistical analysis but text in Section 3.4 doesn't refer t it when comparing different parameters for two experimental groups.

L320-322: Revise the sentence. If no Campylobacter was detected either in control or RFO groups then how did the authors measure the Campylobacter shedding?

L323- What is the limit of detection for C. perfringens in authors' laboratory? <10CFU? Was it determined during the course of these experiments or established previously? Please provide a reference or explanation of how the value was determined.

L392-393: What were the CFUs for Lactobacillus and Bifidobacteria on the day of hatching in control and RFO groups. The data presented in Table 6 does not support this assertion.

L400-406: "The increase of serum ........ RFO injected chickens," this segment seems to suggest that resistance and immune responses in RFO injected chickens will be less pronounced than control group. Please revise the discussion to clearly reflect both the pros and cons of RFO treatment and potential implications.

L471-476: "According ....... IFN-γ." This appears to be highly speculative. Feeding of non-starch oligosaccharides provides the substrate for Lactobacillus and Bifidobacteria, which then produce SCFA. In this study, RFO is injected in eggs and effect s seen after 21 and 42 days old birds. Please revise.

Author Response

Responses to Reviewer 2 Comments (in green)

Dear Reviewer,

Thank You very much for the evaluation of the manuscript. Please find below the detailed answers to the comments as well as the amendments in the revised version which are proposed for Your further consideration:

Point 1.

The Discussion section as presented often is limited to presenting summaries of previous studies and lacks a clear and coherent discussion on the interpretation of the results. For example, section 4.1 presents 3 summaries of three previous studies but does not attempt to make a clear connection to the results of these studies. Another issue is there was no data in Table 4.

Response 1.

Thank You for the comment. Unfortunately, there is limited data about in ovo studies involving prebiotics, provided by other authors. We have intended to address the most relevant information available for RFO, or prebiotic in ovo treatments, keeping in mind the scope of this manuscript. We refer to Berrocoso et al. 2017 [23] and Pilarski et al. 2005 [24] also in the further paragraphs in the discussion.

Moreover:

- we added a paragraph discussing the increased oxidation in meat (Section 4.1., L 390-396)

-amended the paragraph discussing biochemical profile (Section 4.3., L 445-449)

-and proposed an additional paragraph discussing the potential mechanism of RFO action (Section 4.4., L 520- 536).

Point 2.

Table 4 does not have any data.

Response 2.

Unfortunately, a formatting error must have occurred. There should be no quantitative data provided in Table 4. The table was created to indicate the beneficial effects of in ovo delivered RFO in this study. A detailed explanation is also given in Response 8.

Point 3.

L63:  Define FCR at first occurrence.

Response 3.

The definition is provided in the revised manuscript.

Point 4.

L144:  remove space in "cry  pt"

Response 4.

The sentence is corrected in the revised manuscript.

Point 5.

L237:  Correct spelling "Tuckey" to "Tukey"

Response 5.

The spelling is corrected in the revised manuscript.

Point 6.

L302:  The sentence "At day 42, the levels ........  group." is unclear. This unclear sentence is removed from the revised version of the manuscript.  Please clarify if the protein levels in RFO group were decreased compared to control or some other point of reference.

Response 6.

The description of changes in protein levels is rephrased in the amended manuscript as follows:

L313-318,

As a general trend in both groups, a gradual increase in the protein, albumin and globulin levels was observed across the rearing period. Those differences were more pronounced in the Control group (Table 5). At day 1 of rearing, the level of albumin was significantly elevated after the RFO prebiotic treatment compared to the Control group (p=0.033). At day 42, the levels of serum proteins were significantly lower in the RFO group compared to the Control (p=0.002).

Point 7.

L306 -310:  Please revise the paragraph. The sentences, as written, are conflicting and their general meaning is unclear.

Response 7.

The paragraph is revised as follows:

L 322-335,

As a general trend for the lipid profile, a gradual decrease of cholesterol (CHL), high-density lipoprotein (HDL) and low-density lipoprotein (LDL) was observed in both groups from 1 to 42 day (Table 5). For most of the lipid parameters, the values tended to be numerically lower in the RFO group compared to the Control, except for HDL and LDL (day 1) as well as for CHL and HDL (day 42). The level of a beneficial cholesterol fraction (HDL) was significantly higher in the RFO treated chickens compared to the Control group at the end of rearing (p=0.047). The positive changes observed in biochemical profile following the in ovo RFO treatment are additionally indicated in Table 4.

As shown in Table 5, the level of bilirubin was found lower in the RFO group at days 1 (p=0.018) and 42 (p=0.006), and the other hepatic indicators aspartate aminotransferase (AST) (p=0.002) and alanine aminotransferase (ALT) (p=0.001) were significantly lower in the RFO injected chickens at the end of rearing (at 42 days).

At day 42 the serum Ca significantly increased (p= 0.032) in the chickens that were in ovo injected with RFO.

Point 8.

Section 3.4: Was statistical analysis was conducted on blood parameters presented in Table 4 (My version does not have any data in Table 4) and Section 3.4. Table 5 presents statistical analysis but text in Section 3.4 doesn't refer t it when comparing different parameters for two experimental groups.

Response 8.

Table 4 was created to qualitatively highlight the observed beneficial effects of the in ovo injected RFO prebiotic. The statistical significance of these data can be found in the other Tables (3, 5, 6 and 7). In the original draft of a manuscript, there was a ‘tick’ symbol used for the beneficial effects in Table 4 (using the Windings font). For some reason, this formatting was automatically changed to ‘et’ in the pdf of the submitted manuscript and disappeared in the Word version of the submitted manuscript. The ‘+’ sign is provided in the revised manuscript using the acceptable font type and the explanation is given in the table footnote.

Point 9.

L320-322: Revise the sentence. If no Campylobacter was detected either in control or RFO groups then how did the authors measure the Campylobacter shedding?

Response 9.

The sentence is corrected in the revised manuscript:

L 351-353,

For pathogenic species, both groups had negligible Campylobacter levels on day 21 of growth. The Campylobacter level was not reduced in the RFO group at the end of the growing period.

Point 10.

L323- What is the limit of detection for C. perfringens in authors' laboratory? <10CFU? Was it determined during the course of these experiments or established previously? Please provide a reference or explanation of how the value was determined.

Response 10.

The author’s laboratory is accredited to operate under PN-EN ISO/IEC 17025:2005 Standard entitled: "General requirements for research and testing laboratories". The laboratory operating standards assume the number of counted C. perfringens colonies not exceeding 10CFU from horizontal environmental samples, as negligible. The protocol has been applied routinely. Ref. can be found on https://www.iso.org/obp/ui/#iso:std:iso:7937:ed-3:v1:en; based on Schulten S.M., Benschop E., Nagelkerke N.J.D. and Mooijman K.A. Validation of Microbiological methods: Enumeration of Clostridium perfringens according to ISO 7937 (second edition, 1997). Report 286555002, National Institute of Public Health and the Environment, Bilthoven, The Netherlands, 2001 (from page 38. The blank samples are represented by <10CFU/g).

Point 11.

L392-393: What were the CFUs for Lactobacillus and Bifidobacteria on the day of hatching in control and RFO groups. The data presented in Table 6 does not support this assertion.

Response 11.

We have not included a bacteriological profile from the hatched chickens in this study. The study aimed to check how in ovo injected broilers coped with major potential opportunistic pathogens at the further time points of the growing period. Those pathogenic groups do not pose a problem at hatch.

Indeed, we included Lactobacilli and Bifidobacteria counts on days 21 and 42 to widen the profile of major microbiota groups that may be modulated by the prebiotic treatment. Generally, we would have expected an increased number of Lactobacilli and Bifidobacteria in the hatchlings compared to the control groups at hatch, but that difference diminishes in the course of rearing. Below, some data extracted from our previous studies is given:

1/ Previously, we found the increased numbers of Lactobacilli and Bifidobacteria in the hatched chickens injected with other prebiotics (commercial Bi2tos -GOS and DiNovo- algae extract).

Ref. (not cited in the manuscript): Bednarczyk, M., Stadnicka, K., Kozłowska, I., Abiuso, C., Tavaniello, S., Dankowiakowska, A., … Maiorano, G. Influence of different prebiotics and mode of their administration on broiler chicken performance. Animal, 2016, 10(08), 1271–1279. doi:10.1017/s1751731116000173

2/ Trials to optimize the time point for in ovo injection of prebiotics were conducted. We observed the increased number of Bifidobacteria in the hatched chickens that were injected with RFO exactly on the 12 day, compared to the hatchlings that were injected on the other days of egg incubation. These trials are also cited in the discussion.

[15] Villaluenga, C.M.; Wardenska, M.; Pilarski, R.; Bednarczyk, M.; Gulewicz, K. Utilization of the chicken embryo model for assessment of biological activity of different oligosaccharides. Folia Biol. (Krakow) 2004, 52, 135–142.

[24] Pilarski, R.; Bednarczyk, M.; Lisowski, M.; Rutkowski, A.; Bernacki, Z.; Wardeńska, M.; Gulewicz, K. Assessment of the effect of galactosides injected during embryogenesis on selected chicken traits. Folia Biol. (Kraków) 2005, 53, 13–20

Point 12.

L400-406: "The increase of serum ........ RFO injected chickens," this segment seems to suggest that resistance and immune responses in RFO injected chickens will be less pronounced than control group. Please revise the discussion to clearly reflect both the pros and cons of RFO treatment and potential implications.

Response 12.

While interpreting the blood biochemical results, we tend to keep in mind that the blood parameters are highly sensitive to multiple factors and prone to fluctuations among the individuals. The discrepancies in the interpretation of the same parameters can be found across many studies. In this study, we supported our interpretation of physiological indices with other results - the observed optimal performance in both groups, but a healthier morphology of the gut and apparently better combating the potential opportunistic pathogens during the rearing, after the in ovo prebiotic treatment.

The following fragment is additionally provided in the revised manuscript:

L 445-449,

‘However, we observed a lower protein level in the serum of RFO-injected birds compared to the Control group at day 42 of rearing. This might result from a good metabolic turnover in the RFO treated birds, which allowed for better utilization of the nutrients and a lower requirement for protein synthesis at the end of fattening, thus helping to maintain the metabolic balance during the period of rapid growth (Tóthová et al., 2019 [33]).’

Point 13.

L471-476: "According ....... IFN-γ." This appears to be highly speculative. Feeding of non-starch oligosaccharides provides the substrate for Lactobacillus and Bifidobacteria, which then produce SCFA. In this study, RFO is injected in eggs and effect s seen after 21 and 42 days old birds. Please revise.

Response 13.

Thank You for this comment. We fully agree that the status of knowledge might not allow to conclude about the mechanism of action of RFO against C. perfringens. Therefore, we plan to perform further studies to elucidate the mode of action of prebiotics and probiotics by using in ovo model and in vitro/in vivo studies involving metagenomic analyses, metabolic and gene expression patterns to elucidate the mechanisms of action of bioactive compounds.

The reference that we cite in the manuscript (Teng et al., 2018, [6]) refers to fructooligosaccharides, which, alike RFO and GOS, fall in the group of non-starch oligosaccharides. A single dose of GOS injected in ovo [Angwech et al. 2019, [10]) had an effect on the reduction of Eimeria oocysts in kuroiler chickens, long after in ovo treatment (after 18 weeks).

To support the suggestion given in discussion, we proposed to add the following paragraph at the end of Section 4.5:

L 520-536,

‘Furthermore, our previous studies allow suggesting that in ovo injected RFO prebiotic may influence the microbiota composition and pose immunomodulatory effects later in chicken life. Previously, a single injection in ovo of a synbiotic based on RFO differentially modulated immune-related genes expression; IL-12 gene expression was silenced after 3 and 6 weeks of rearing in Cobb broilers (Dunislawska et al., 2017 [52]). In ovo injection of RFO prebiotic alone or in a synbiotic combination with Lactobacillus plantarum had an effect on immune system development after 3 and 6 weeks of rearing. The bursa development was faster, bursa to spleen index was higher and the increased number of developing T-cell progenitors in thymus was observed (Slawinska et al. 2014 [53]). A range of other life-long phenotypic effects after in ovo delivery of RFO compound is also listed in Table 1 in Siwek et al. (2018, [7]).

Based solely on the research committed to in ovo injection of RFO, the following mechanism of action of RFO can be suggested:

1/competitive reduction of access to colonization sites for potential opportunistic pathogens by promoting the proliferation of Bifidobacteria at hatch [15,24]

2/ improving the ileum mucosa morphology and immunity in the small intestine [8, 23]

3/stimulation of the immune system by accelerated development of lymphatic organs and impact on cytokines genes expression in spleen and cecal tonsils [52,53]

In the case of adding that paragraph, 2 additional references must have been provided in the revised manuscript:

[52] Dunislawska, A.; Slawinska, A.; Stadnicka, K.; Bednarczyk, M.; Gulewicz, P.; Jozefiak, D.; Siwek, M. Synbiotics for Broiler Chickens-In Vitro Design and Evaluation of the Influence on Host and Selected Microbiota Populations following In Ovo Delivery. PloS one 2017, 12, e0168587.

[53] Slawinska, A.; Siwek, M.; Bednarczyk, M. Synbiotics injected in ovo regulate immune-related gene expression signatures in chicken. Am. J. Vet. Res. 2014, 75, 997–1003.

Reviewer 3 Report

The paper presented for review brings new elements to the current state of knowledge regarding the effect of oligosaccharides from the raffinose family on the 12th day of incubation of the egg modulates intestinal development and resistance to opportunistic pathogens in broiler chickens. The work has great research potential and very high practical significance. RFO obtained from local legumes can be considered a promising prebiotic for broiler chickens. Because in ovo RFO injection can very early stimulate chicken immunity.The purpose of the work is clearly stated. The conclusions of the conducted research are clear and result from the obtained research results. The material used for the tests is sufficient, the test methods have been selected accordingly.
The layout of the tables and graphs is correct. The differences between the groups were marked correctly. Only Table 4 is not understood. There were no numerical values of health indicators on days 21, 35 and 42 on chicken rearing. Discussing the results against the background of other authors is very detailed. The publications cited by the authors of the article are well selected. For the most part, the authors refer to the latest knowledge published in renowned scientific journals.

Author Response

Responses to Reviewer 3 Comments (in blue)

Dear Reviewer,

Thank You very much for the evaluation of the manuscript. Your point is addressed below. The manuscript was carefully revised for further consideration.

Point 1. 

Only Table 4 is not understood. There were no numerical values of health indicators on days 21, 35 and 42 on chicken rearing. 

Response 1.

Table 4 was created to qualitatively highlight the observed beneficial effects of the in ovo injected RFO prebiotic. The values of health indicators and statistical significance of these data can be found in the other Tables (3, 5, 6 and 7).

In the originally submitted manuscript, there was a ‘tick’ symbol used for the beneficial effects in Table 4 (using the Windings font). For some reason, this formatting was automatically changed to ‘et’ in the pdf of the submitted manuscript and disappeared in the Word version of the submitted manuscript. The ‘+’ sign is provided in the revised manuscript using the acceptable font type and the explanation is given in the table footnote.

Round 2

Reviewer 1 Report

No further comments.